# Use of Ozone Therapy in Diabetic Foot Ulcers

**DOI:** 10.3390/jpm13101439

**Published:** 2023-09-27

**Authors:** Álvaro Astasio-Picado, Alba Ángel Babiano, Miriam López-Sánchez, Rocio Ruiz Lozano, Paula Cobos-Moreno, Beatriz Gómez-Martín

**Affiliations:** 1Physiotherapy, Nursing and Physiology Department, Faculty of Health Sciences, University of Castilla-La Mancha, 45600 Toledo, Spain; albaangelb01@gmail.com; 2Extremadura Health Service, 10300 Cáceres, Spain; miri-due@hotmail.es (M.L.-S.); rociorulo@gmail.com (R.R.L.); 3Nursing Department, University of Extremadura, Plasencia (Cáceres), 10600 Plasencia, Spain; pacobosm@unex.es (P.C.-M.); bgm@unex.es (B.G.-M.)

**Keywords:** diabetic foot, ozone therapy, diabetic foot ulcer

## Abstract

Introduction: ozone therapy is a therapy composed of ozone. This gas is in the atmosphere with various general effects: direct disinfectant and trophic effects and a systemic antibacterial and antiviral effect. This gas also improves blood circulation, makes glucose metabolism more effective, improves erythrocyte metabolism, and improves fatty acid metabolism. Objective: Provide evidence of the effectiveness of ozone therapy in wounds of patients with diabetic foot. Analyze the effectiveness of ozone therapy compared to other treatments to achieve good wound healing in patients with diabetic foot. To study the benefits of the use of ozone therapy in ulcers of patients. Analyze the management of ozone therapy and other treatments to achieve healing of ulcers in patients. Methodology: A bibliographic review focused on articles published between November 2014 and June 2023 was carried out. The following databases were consulted: Pubmed (Medline), Dialnet, Google Scholar, Web of Science (WOS), Scielo, and Scopus. Results: After applying the article selection criteria and evaluating the quality of the methodology, a total of 17 articles were obtained. The results affirm ozone therapy as promising for the treatment of wounds in patients with diabetic foot. Conclusions: the evidence has been able to determine that ozone therapy is adequate for the treatment of diabetic foot ulcers. In addition, the therapy has been shown to be effective, safe, and beneficial, with few adverse effects for the treatment of diabetic foot ulcers.

## 1. Introduction

Diabetic foot (PD) is a chronic complication of diabetes mellitus (DM), whose factors are ischemia, infection, poor metabolic control, and neuropathy. Ulcers, gangrene, and amputations appear in the lower extremities of patients with an at-risk foot. This can lead to partial or permanent disability in the patient [1]. Patients with diabetic foot should be admitted to specific units with strategic, permanent, and multidisciplinary care plans, which include treatment and prevention [1]. The diabetic foot must be approached from a multidisciplinary point of view since this problem is highly prevalent, and its complications require the participation of various professionals, such as nursing, orthopedics, podiatry, and medicine [1,2].

### 1.1. Concept, Diagnosis, and Classification of the Diabetic Foot

Diabetic foot is a chronic complication of diabetes mellitus, whose etiology is multifactorial. Its main factors are infection, ischemia, neuropathy, and poor metabolic control. Ulcers, gangrene, and amputations in the lower limbs are frequent clinical characteristics which can cause the patient to be partially or permanently disabled [1,2]. 

It is important to make a correct diagnosis in order to be able to detect the signs and symptoms and to be able to prevent disability and limb loss [2]. To make a correct diagnosis, the anamnesis, physical, neurological, and vascular examination and complementary tests must be evaluated [3]. 

The scales most used to classify the diabetic foot are the Wagner scale, the UT scale (University of Texas), and the scale of the international PEDIS group. The most accepted, known, and simple scale in the world is the Wagner scale. The UT scale classifies the depth of the wound, infection, and peripheral arterial occlusive disease [4]. These categories must be combined with the depth of the ulcer following the Wagner classification [1]. The PEDIS classification evaluates five parameters, these being the most important in the investigation of diabetic ulcers. These categories are evaluated independently [5].

### 1.2. Diabetic Foot Treatment

The diabetic foot ulcer should be cleaned with physiological saline. The use of chlorhexidine, povidone iodine, or hydrogen peroxide is not recommended because it can produce a cytotoxic effect. In the event that the ulcer is necrotic, a debridement is performed in order to eliminate the damaged tissue. Afterwards, the use of synthetic dressings is recommended, which helps to absorb the exudate and keep the ulcer moist, helping its healing [3]. 

After diagnosis, it is initially treated with empirical antibiotic therapy, which can be modified according to the patient’s symptoms and improvement. Before starting treatment, it should be taken into account if the patient presents any resistance to antibiotics and the degree of infection that they present. If the presence of anaerobic or Gram bacteria is anticipated, broad-spectrum antibiotic therapy must be used, with efficacy against anaerobic pathogens [3]. 

Surgical treatment options include debridement and revascularization. Debridement: This treatment is used to achieve adequate healing. Necrotic tissues and bones or fragments that are compromised should be debrided. Normally, lesions appear in the nails, and their removal is necessary in order to drain the purulent secretion. Interdigital and digital lesions can also be observed; depending on their injury, they will need rest, antibiotics, or debridement. The patient’s wound should be constantly observed to avoid complications or future amputations [3]. Revascularization: This treatment is usually used when the patient is unable to perform activities of daily living due to the symptoms they present. The illustration of performing angioplasty is provided by tests such as angiography and Doppler [3].

### 1.3. Ozone Therapy

Ozone is a gas present in the stratosphere whose decomposition rate is high. Ozone is produced by three major sources of energy: electrical discharges, chemical electrolysis, and ultraviolet light radiation [5,6]. 

Ozone presents properties capable of reacting with inorganic and organic substances until the formation of carbon oxides, higher oxides, and water; that is, until its complete oxidation [6]. 

The general effects of this gas are: when applied locally, it has direct disinfectant and trophic effects; thanks to the formation of peroxides, it has a systemic antibacterial and antiviral effect; it increases the deformity of red blood cells, thus improving blood circulation; it favors the delivery of oxygen to the tissues; the metabolism of glucose becomes more effective, improving the metabolism of erythrocytes; finally, thanks to the activation of antioxidant enzymes that are responsible for eliminating peroxides and free radicals, there is an improvement in the metabolism of fatty acids [6]. 

The metabolic effects that can be highlighted are: increased use of glucose at the cellular level, improvement in protein metabolism, and a direct effect on unsaturated lipids, which is responsible for oxidizing it and inducing repair mechanisms at the same time [6]. 

Ozone has various adverse effects; however, it is not a drug, so there are no allergic reactions or interactions with other drugs. When its administration is carried out with high doses, the patient may present a feeling of heaviness. This usually occurs in a few patients, with a short duration and rapid resolution. Rarely, the painful stimulus due to the puncture of the needle can cause a vagal crisis in the patient without the need to use pharmacotherapy to reverse it [6]. 

Patients in whom the use of this therapy will be contraindicated will be patients with a large glucose six phosphate dehydrogenase deficiency since red blood cells may oxidize and cause hemolysis. It should not be performed in patients with thrombocytopenia and hyperthyroidism either. It should also not be performed in those with severe cardiovascular instability, hemorrhagic pictures, or convulsive status [6].

The objective of this study is to analyze the effectiveness of ozone therapy in wounds of patients with diabetic foot, analyze the effectiveness of ozone therapy compared to other treatments to achieve good wound healing in patients with diabetic foot, to study the benefits of the use of ozone therapy in ulcers of patients, and analyze the management of ozone therapy and other treatments to achieve healing of ulcers in patients.

## 2. Materials and Methods

The preparation of this work was carried out through a systematic bibliographic review of the articles found by searching the following databases: Medline/Pubmed, Dialnet, WOS, Scielo, Scopus, and Google Scholar. To determine the best possible scientific evidence, a series of inclusion and exclusion criteria were applied.

### 2.1. Information Sources and Search Strategy

The keywords for this review are: diabetic foot, ozone therapy, and diabetic foot ulcer. To carry out the bibliographic search, different keywords in English were used, such as: “diabetic foot”, “ozone therapy”, “ozone therapy”, “diabetic foot”, “ozone therapy”, and “ulcers”. These were validated by DeCS and MeSH. Once selected, the corresponding Boolean operators were used: AND, as well as the necessary parentheses and quotation marks. The final search string was as follows: ((Diabetic foot) AND (Ozone therapy)). 

### 2.2. Inclusion Criteria and Exclusion Criteria

The criteria that were taken into account for the selection of the relevant studies were the following. Inclusion criteria: the period between 2014 and 2023; article type: article review and article research; field: medicine; English language; sample in the adult population; studies that provide scientific evidence justified by the level of indexing of articles in journals according to the latest certainties. Exclusion criteria: articles prior to 2014; language: not English; studies in which the population included minors; studies that do not provide scientific evidence justified by the level of indexing of articles in journals according to the latest certainties.

### 2.3. Methodological Evaluation of the Data Used

For the methodological evaluation of the individual studies and the detection of possible biases, the evaluation was carried out using the PEDro Evaluation Scale. This scale consists of 11 items, providing one point for each element that is fulfilled. The articles that obtained a score of 9–10 points have excellent quality, those between 6 and 8 points have good quality, those that obtained 4–5 points have intermediate quality, and, finally, those articles that obtained less than 4 points have poor methodological quality article.

The Scottish Intercollegiate Guidelines Network classification was used in the data analysis and assessment of the levels of evidence, which focused on the quantitative analysis of systematic reviews and the reduction in systematic error. Although it took into account the quality of the methodology, it did not assess the scientific or technological reality of the recommendations.

## 3. Results

The research question was constructed following the PICO format (population/patient, intervention, comparator, and outcomes/outcomes), detailed as P (patients): adult subjects of both sexes; I (intervention): use of ozone therapy in patients with diabetic foot ulcers; C (comparison): traditional treatments for ulcers in patients with diabetic foot; O (results): effectiveness of ozone therapy (Figure 1).

Below is a table that shows the search strategy used to select the 17 articles selected from the six databases, following the criteria of identified studies, duplicate studies, title, abstract, full text, and valid studies of a definitive nature. The total number of valid articles is summarized in Appendix A.

In the studies by Suchin Dhamnaskar et al., Jing Zhang et al., Jaróslaw Pasek et al., Xiaoxiao Hu et al., and Laura Gheuca Solovastru et al., no change in wound area was observed at the beginning of treatment (11.742% versus 10. 82%), but 21 days after ozone administration, a reduction in the wound area of 18.62% could be seen; with statistical significance, its values were <0.001 and 0.022, respectively (Table 1) [7,8,9,10,11].

In the study by Jaróslaw Pasek et al., a reduction in area was observed, taking into account sex, age, BMI (body mass index), duration of the disease, and location of the ulcer (Table 2) [9].

After the administration of ozone treatment, there was a reduction in the wound area of 38.74%. Fifty-four patients participated in the study; 2 of the participants healed completely (3.7%), 18 of them achieved a reduction greater than 50% of the initial value (33.3%), and the remaining 34 patients observed a smaller reduction of 50% of the initial value (63.0%) [9]. 

In the study by Jaróslaw Pasek et al., they observed a 60% reduction in pain. In 2 of the patients, a complete reduction was achieved (3.7%), in 48 patients, the pain was greater than 50% of the baseline pain (92.5%), and in 4 patients, the pain intensity was reduced but did not exceed 50% of baseline (7.4%) [9]. 

In the articles by Yi-Ting Zhou et al. and Mufarika et al., healing was taken into account, being greater in patients treated with ozone therapy. After 12 months of follow-up of this treatment, the patients who were treated with ozone obtained a healing of 92%, while the rest of the patients treated with standard treatment obtained a healing of 76.19% in the first study. In the second study, the mean value before the test was 42.77 and after the test 20.69 after ozone application [12,13]. In the trial by Morteza Izadi et al., they also discussed the healing time, being 69.44 ± 36,055 days in the ozone group, while in the control group, after 180 days, 25% of the patients did not heal completely [14]. 

In the study by Suchin Dhamnaskar et al., they observed that patients treated with ozone had 20% more granulation tissue than conventionally treated patients [7]. In the study conducted by Myroslav V. Rosul et al., a reduction in the burning sensation in the foot, a constant cooling of the feet, and paresthesia could be observed [15].

Erin Fitzpatrick et al. carried out a systematic review where they reviewed various trials in which different results could be observed, as well as different age ranges and duration of treatment (Table 3) [16]:

Kasmawati Kadir et al. carried out a test where they observed a significant reduction in the number of bacteria in the group treated with ozone (*p* = 0.001), while in the control group, there was no reduction (*p* = 0.06). The number of bacteria was higher before treatment (*p* = 0.334) than after it (*p* = 0.037) [17]. 

Navid Faraji et al. carried out a case report on a patient who underwent several ozone therapy sessions for 30 days. After the sixth session, the deep parts of the wound were closed as the granulation tissue grew rapidly. After one month of treatment, the ulcer had completely closed [18]. 

A systematic review was carried out by Andressa Urbano Machado et al., where they were able to compare various trials and obtain different results (Table 4) [19].

Lima Bomfim et al. carried out a systematic review, where different results were observed in various trials in which they used ozone for the treatment of wounds (Table 5) [20].

Gheuca Solovastru et al. carried out a study where changes in the mean speed of healing could be observed. In the first 7 days, it was reduced by 0.17 cm^2^, 0.14 cm^2^ between days 7 and 14, and 0.04 cm^2^ in the last days of treatment [11]. Aparecida Oliveira Modena et al. carried out a systematic review, which showed several trials with different results in each one (Table 6) [21].

Quing Wen et al. and Svitlana Y. Karatieieva et al. carried out studies where various results could be observed, highlighting an improvement in healing and a decrease in the amputation rate and wound area [22,23]. 

Jing Zhang et al. conducted a study where various changes in growth factors could be seen. They compared the ozone group with the control group. At the beginning of the treatment, they could not observe major differences in vascular endothelial growth factor (VEGF), with 19.95% in the ozone group and 17.93% in the control group. After administration of ozone treatment, the results increased to 34.86% in the ozone group and 26.44% in the control group [8]. 

They also studied transforming growth factor (TGF). When ozone was administered, the values of the ozone group were 4.48%, and those of the control group were 5.17%. After its administration, there was an increase in these values, that of the ozone group being 14.95% and the control group being 10.45% [8]. 

They also looked at the results for platelet-derived growth factor (PDGF) proteins. When ozone treatment was administered, the values were 14.23% in the ozone group and 15.50% in the control group. After its administration, the values were 31.44% in the ozone group and 20.78% in the control group [8].

## 4. Discussion

The studies reviewed in this research paper provide information on ozone therapy in wounds of patients with diabetic foot. 

Several authors discuss the reduction in the amputation rate thanks to the use of ozone therapy. Morteza Izadi et al. determined a decrease in C-reactive protein and in the erythrocyte sedimentation rate; by reducing this, the amputation rate was also reduced [14]. Lima Bomfim et al. demonstrated that ozone was an effective adjuvant therapy, helping to prevent complications in the disease, as well as amputations. In addition, to improve the results they obtained, they wanted to carry out evidence-based protocols for ozone treatment [20]. Quing Wen et al. stated that the quality of the studies they included was poor; despite this, thanks to the use of ozone, both individually and in combination, they were able to observe a reduction in the amputation rate [22]. 

According to Suchin Dhamnaskar et al., the use of ozone accelerated the formation of granulation tissue. After 21 days of treatment administration, wound healing was 27% faster in the ozone group, and signs of inflammation disappeared [7]. Myroslav V. Rosul et al. used ozone both locally and systemically, which improved the appearance of granulation tissue [15]. 

Regarding the reduction in the wound area, Quing Wen et al. observed a decrease in this but did not contemplate an improvement in the proportion of wounds completely healed [22]. Gheuca Solovastru et al. perceived a significant and progressive reduction in the wound, being superior to the control group, which did not improve after the observation period [11]. On the other hand, in the study by Yi-Ting Zhou et al., no adverse effect was observed [12]. Jaróslaw Pasek et al. considered the decrease in area, age, sex, body mass index, duration of the disease, and location of ulcers [9]. 

Regarding hospital stay, Suchin Dhamnaska et al., Lima Bomfim et al., and Myroslav V. Rosul et al. observed a reduction in this in each study. In the study carried out by Myroslav V. Rosul, the hospitalization time of the ozone group was 17.09 days [15]. However, in the study carried out by Suchin Dhamnaska et al., it was 9 days. By reducing hospitalization time, overall cost expenditure was indirectly reduced [7]. Lima Bomfim et al. observed an improvement in wound healing and, as a consequence, a reduction in the stay of patients in hospital [20].

To evaluate the extent of pain, both Jaróslaw Pasek et al. and Xiaoxiao et al. used the visual analogue scale (VAS). To evaluate pain, Jaróslaw Pasek et al. took into account age, sex, body mass index, duration of the disease, and location of the ulcer. In this study, a reduction in pain could be observed, although it was reduced more in some patients than in others [9]. On the other hand, although Xiaoxiao et al. also experienced pain reduction, they admitted that this treatment needs more research since the sample was small and all cases were performed at the same center [10]. Regarding the use of VAS, a statistically significant reduction in pain intensity on the VAS scale was evident in all patients, median of 6 (5–7) points before treatment versus 4.4 (3–7) points after treatment, *p* = 0.000001) [9]. Similarly, treatment duration was significantly shorter and dressing change times and maximum VAS scores were dramatically lower in the combined group than in the VAC group [10].

Ozone therapy hastened the healing process in several studies. According to Morteza Izadi et al., ozone decreases the FBS (fasting blood sugar) level, improving wound healing [14]. In the study by Qing Wen et al., healing in the ozone group was not much higher than in the control group 12 months after treatment, so the quality shown in this study was low [22]. According to Suchin Dhamnaskar et al., the number of ozone sessions administered and healing could have a certain relationship, but more studies are needed to prove it [7]. Jing Zhang et al. commented that diabetic foot ulcer healing was improved after control of platelet-derived growth factors and transforming growth factors [8]. According to Gheuca Solovastru et al., from practice, healing depends on various factors, such as the preparation of the wound bed [11]. Mufarika et al. observed healing acceleration thanks to the use of ozone on wounds compared to patients who were not administered this [13]. 

Navid Faraji et al. carried out a case report after ozone administration, and an acceleration in wound healing was observed, thus improving the patient’s quality of life [18]. On the other hand, although Kasmawati et al. did not perceive changes in wound healing, she was able to see how changes were made in the bacteria presenting it, reducing their colonies [17]. 

Fitzpatrick et al., Urbano Machado et al., and Aparecida Oliveira Modena et al. conducted a systematic review. According to Fitzpatrick et al., thanks to their meta-analysis, they saw that there is good evidence of the use of ozone, in addition to being effective, but more research is needed on the matter [16]. While Aparecida Oliveira et al. studied its short-term efficacy, more studies are needed to determine what the adverse effects are [21]. Urbano Machado et al. perceived several adverse effects; despite this, they concluded that ozone can be used as a complementary treatment [19].

In the same way, in the study carried out by Svitlana Y. Karatieieva, she saw that ozone did not negatively affect the levels of intoxication of the body or homeostasis; in addition, it improved the evolution of the wound in purulent processes [23].

## 5. Conclusions

In relation to the main objective of demonstrating the effectiveness of ozone therapy in wounds of patients with diabetic foot, it was verified that the use of ozone therapy has positive effects on wounds of patients with diabetic foot, accelerating their healing and scarring, in addition to improving the quality of life of the patient. This refers to the objective of analyzing the effectiveness of ozone therapy compared to other treatments to achieve good wound healing in patients with diabetic foot. It has been shown that ozone therapy positively affects the wounds presented by patients with diabetic foot. Its use, compared to other treatments, accelerates wound healing, which leads to a reduction in cost and hospital stay. Despite this, little research has been conducted on ozone therapy, so more research is needed on this therapy. Regarding the objective of studying the benefits of the use of ozone therapy in ulcer patients, great results have been obtained. It was evidenced that it is an effective, safe, and beneficial therapy, and few adverse effects have been observed. The wounds presented better healing, a reduction in their area, a decrease in bacterial colonization, and a reduction in the amputation rate, reducing hospital stay and, consequently, reducing cost, as mentioned above. Finally, the objective is to analyze the management of ozone therapy and other treatments to achieve the healing of ulcers in patients. It was concluded that the use of a standard treatment and ozone therapy simultaneously has beneficial effects, accelerating the complete healing of the wound.

## Data Availability

Not applicable.

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
