# Peer review of "Use of Ozone Therapy in Diabetic Foot Ulcers"

_jpm, 2023, doi:10.3390/jpm13101439_

Round 1
Reviewer 1 Report
In this manuscript the authors have conducted a review on the use of ozone therapy on diabetic foot ulcers. Generally the review is well written, however it can be improved by addressing the following:
1. The review has a repetitive nature, often mentioning the same results in different contexts. This can make the manuscript appear redundant and may cause the reader to lose interest.
2. While the manuscript covers a wide range of studies, it seems to gloss over negative or neutral findings, which could give a somewhat biased viewpoint. For instance, the study by Quing Wen et al, which reported poor quality and inconclusive results on ozone therapy's effectiveness, is mentioned but not delved into deeply.
There are inconsistencies in the manuscript. For instance, Gheuca Solovastru et al and Yi-Ting Zhou et al are both mentioned under the discussion of wound area reduction with reference number [11]. This could be a typographical error, but it raises questions about the accuracy of other references.
3. Language and Clarity
While the manuscript is largely well-written, there are instances where clarity could be improved. Some sentences appear too long and may benefit from restructuring to ensure that the core message is not lost.
4. Analysis and Interpretation
The manuscript provides a comprehensive overview, but a deeper analysis of the discrepancies between the studies, the reasons for such differences, and any potential conflicts of interest would have strengthened the review.
The Visual Analogue Scale (VAS) used to assess pain is mentioned, but the actual results and the statistical significance, if any, aren't detailed. A discussion about the scale's reliability and validity in the context of these studies would have been insightful.
5. Recommendations
General Conclusions: It would be beneficial if the conclusions section drew more specific insights from the individual studies. While the overarching findings are presented, nuances from individual studies could provide a more robust conclusion.
Suggestions for Future Research: The conclusion rightly points out that more research is needed on ozone therapy, especially given the poor quality of some studies. However, it might be beneficial to suggest specific areas or methodologies for future research.
Clinical Implications: Given the potential benefits and the few adverse effects reported, the manuscript could further examine into the practical applications and recommendations for clinicians.
6. Conclusion
Overall, the manuscript offers a comprehensive review of the use of ozone therapy in treating diabetic foot wounds. While the positive outcomes are promising, a more critical analysis, highlighting both the strengths and limitations of existing studies, would enhance the paper's credibility and provide a balanced viewpoint.
The English requires minor editing
Author Response
Dear reviewer,
We appreciate the review of the manuscript. Your evaluations and contributions will improve its quality.
Regarding your considerations:
Sections 1 and 2. An objective, clear synthesis with the best evidence of all the manuscripts used in this work has been carried out.
Regarding the negative or neutral findings, the manuscript by Quing Wen et al, a reduction in the wound area is observed. Although it is true that wounds take time to heal. All these positive and negative considerations are included in the manuscript. The bibliographic reference numbers have been completely revised to avoid typographical errors.
Section 3. Language and clarity: the manuscript has been revised by synthesizing long sentences into shorter sentences.
Section 4. Analysis and Interpretation:
We have incorporated into the manuscript details of results with statistical significance that give reliability and validity to the visual analogue scale.
Section 5. Recommendations:
General conclusions: we have re-evaluated the same to provide a more solid conclusion.
Suggestions for future research: We greatly appreciate recommendations for future research.
Comments on the quality of the English language:
The quality of English in the relevant sections would need to be improved: translation has been carried out by the University's language department.
We are very grateful for your assessment. We hope that the article is to your liking and acceptance.
Reviewer 2 Report
Thank you for the opportunity to give comments to this review.
Some comments from me:
1) Would suggest for the authors to re-write the introduction as it significantly overlaps with section 1.1 and I would not think using "As I said before" is appropriate in this section.
2) Section 1.1 - I noticed acronyms PD and DM appeared in this section and not in the Introduction. For consistency, acronym should appear in the first time the word appears.
3) Section 1.2 - this section is too long and not the main focus of the study. Would consider to summarise further as the focus is on section 1.3
4) Did the authors register this protocol on PROSPERO or OSF?
5) Section 2.1 - I don't see supplementary files for search strategy for each database and the number of articles obtained from each database. This section seems rather incomplete. Did the study team consult a medical librarian to confirm the search terms? Who did the search?
6) Section 2.2 - why 2014 is the cut off year for this review? why the study team never consider other languages besides English?
7) Section 2.3 - who did the evaluation of the articles? it is important to state this and what happens if there are conflicts?
8) How about data extraction table? How was it developed? Once again this information is missing.
9) Figure 1 - what were the reasons records were removed (n=377)? would be good to state the reasons.
10) I don't see a section of Limitations or Implications of Practice in this review.
I find the tables too cluttered and would suggest to restructure the tables to be precise and concise.
Can consider to only include "last name" at al for all articles instead of 'full name" et al.
Would need to improve the English quality on relevant sections.
Author Response
Dear reviewer,
We appreciate his contributions to our manuscript. These contributions will make the article have greater scientific and methodological validity.
Regarding your comments:
1) I would suggest the authors rewrite the introduction, as it overlaps significantly with section 1.1 and I don't think using "As I said before" is appropriate in this section - the introduction has been changed.
2) Section 1.1: I noticed that the acronyms PD and DM appeared in this section and not in the Introduction. For consistency, the acronym should appear the first time the word appears: modified.
3) Section 1.2: This section is too long and is not the main focus of the study. I would consider summarizing further, as the focus is on section 1.3: we proceed to summarize the last paragraph for its length and content.
4) Did the authors register this protocol in PROSPERO or OSF?: it is a bibliographic and narrative review with many PROSPERO criteria, but we have not proceeded to register it.
5) Section 2.1 - I see no supplementary files for the search strategy for each database and the number of articles obtained from each database. This section seems quite incomplete. Did the study team consult a medical librarian to confirm the search terms? Who did the search?: The databases used were Medline/Pubmed, Dialnet, WOS, Scielo, Scopus and Google Scholar. The keywords of the manuscript were validated by the DeCS and MeSH. The search string was validated by both the research team and the help of the university library staff. A table with the total number of articles for each database was not added since some reviewers considered duplicating the information. We have no problem adding the table.
6) Section 2.2: Why is 2014 the cutoff year for this review? Why does the study team never consider languages other than English?: Two previous searches for manuscripts were carried out and the high quality of scientific works that met that date and that language were seen.
7) Section 2.3 - Who carried out the evaluation of the articles? It is important to state this and what happens if there are conflicts?: The evaluation of the articles was carried out by the research group. In case of conflict, due to the presence of an article authored within the group, an external evaluation would be used. This external evaluation was not accurate at any time.
8) How about the data extraction table? How did it develop? Once again this information is missing: the table is developed as the selection of manuscripts is carried out, answering the PICO question. This document is in section 3.
9) Figure 1: What were the reasons why records were deleted (n=377)? It would be good to explain the reasons: it is indicated in the second paragraph of section 3 "Below is a table that shows the search strategy used to select the 17 articles selected from the 6 databases, following the criteria of studies identified, duplicate studies, title, abstract , full text, and valid studies of a definitive nature".
10) I do not see a Limitations or Practice Implications section in this review: we appreciate this consideration. In the group we had our discussion about whether or not to include this section.
We consider that the article itself is an implication to practice.